# Contemporary Continuous Aggregation: A Robust Categorical Encoding for Zero-Shot Transfer Learning on Tabular Data

## Abstract

Tabular data, as the most fundamental structure of many real-world applications, has been a spotlight of machine learning since the last decade. Regardless of the adopted approaches, e.g., decision trees or neural networks, *Categorical Encoding* is an essential operation for processing raw data into a numeric format so that machine learning algorithms can accept it. One fatal limitation of popular categorical encodings is that they cannot extrapolate to unseen categories for machine learning models without re-training. However, it is common to observe new categories in industry, while re-training is not always possible, e.g., during the cold-start stage with no target examples. In this work, we propose Contemporary Continuous Aggregation (CCA), a novel and theoretically sound categorical encoding which can automatically extrapolate to unseen categories without any training. CCA only relies on statistics from raw input that can be maintained at low time and memory costs, thus it is scalable to heavy workloads in real-time. Besides, we also empirically showcase that CCA outperforms existing encodings on unsupervised unseen category extrapolation, and achieves similar or even better performance in normal situations without extrapolation, promising CCA to be a powerful toolkit for tabular learning.

## 1 Introduction

Tabular data, consisting of a collection of the same combinations of categorical and continuous values, is the most basic and common data type in real-world domains including fraud prevention (Cartella et al., 2021; Khatri et al., 2020), medical profiling (Ogunleye & Wang, 2019; Zhang et al., 2020), molecular analysis (Babajide Mustapha & Saeed, 2016; Bi et al., 2020; Chen et al., 2020), advertising recommendation (Zhang et al., 2019), and anomaly detection (Pang et al., 2021). With the great success of artificial intelligence in recent years, many machine learning approaches have been proposed and adopted to automatically learn to predict from a giant amount of samples (observations). Representative state-of-the-art works include XGBoost (Chen & Guestrin, 2016), CatBoost (Prokhorenkova et al., 2018), and LightGBM (Ke et al., 2017) from the conventional Gradient Boosted Decision Tree (GBDT) family, along with deep neural network competitors such as TabTransformer (Huang et al., 2020), FT-Transformer (Gorishniy et al., 2021), VIME (Yoon et al., 2020), SAINT (Somepalli et al., 2021), and many other recent works (Arik & Pfister, 2020; Popov et al., 2019) that achieve similar or slightly worse performance.

One fundamental requirement of all the aforementioned state-of-the-art approaches, regardless of GBDTs or neural networks, is that the input tabular data must be preprocessed into continuous values. However, real-world tabular data often contains categorical values, thus it is necessary to convert categorical values into continuous values, and such conversion techniques are called *Categorical Encodings* (Hancock & Khoshgoftaar, 2020). The most popular categorical encoding in industry is *CatBoost (target) encoding* for GBDTs, which utilizes correlation information between categories and learning targets, e.g., target label distribution w.r.t. each category; and *parametric encoding* for neural networks, which tunes category encodings as a group of learnable parameters through gradient descent optimization.

For most existing categorical encodings, they assume that all categories are known during learning, and when an unseen category appears at inference time, machine learning approaches will treat it as an "Unknown" category, i.e., a preset category for unseen categories (Pargent et al., 2019). While this trick works well for stable systems whose unseen categories are rare, it will not work well for highly dynamic systems. For instance, in social media or video streaming, "tags" is an important categorical variable for recommendation, whose cardinality increases frequently over time. Simply treating all new tags as an unknown category may lead to poor service quality since up-to-date recommendations are often related to those new tags. The most common industrial solution is to periodically re-train new category encodings and machine learning models, thus covering new categories (Bifet & Gavalda, 2007; Gama et al., 2014). However, training and deploying new models, especially for large service systems, will cost a lot of resources (Le & Hua, 2021), and new categories will still be weakly handled between two training periods.

Another critical industry challenge is that learning targets for training new models are generated and collected with huge delays. For example, in a fraud detection system, it may take investigators weeks or months to identify only tens of fraudsters, while hundreds or thousands of labeled samples are at least required for re-training. Thus, it is impossible to train new models to keep pace with category increments. Another challenge is *Category Shifting* in transfer learning: Suppose we already have a machine learning model tuned by data from one region, and plan to initialize the service in a new region. The machine learning model should work directly since the task stays the same. However, categories can possibly be expressed in different text, merged into new categories, or split into sub-categories in a different region. Therefore, the machine learning model fails drastically in practice since its encoding system cannot handle such shifting and treats most categories as "Unknown".

**Contributions.** In this work, we propose **Contemporary Continuous Aggregation (CCA)**, a novel categorical encoding to address the zero-shot transfer learning challenge with category shifting and increasing category cardinality for highly dynamic applications:

- We first formalize the requirements and expressivity of categorical encodings, providing a theoretical analysis toolkit for researchers to justify the power of any categorical encoding design.
- We then implement CCA, a scalable and theoretically sound categorical encoding with the ability to extrapolate to unseen categories. Our experiments show that CCA outperforms other encodings on unsupervised extrapolation, and performs similarly or even better on tasks without extrapolation.

## 2 PRELIMINARIES

In general, a collection of tabular data can be defined as a pair of categorical and continuous data, $\mathbf{X} = (\mathbf{C}, \mathbf{N})$, where $\mathbf{C} \in \mathbb{N}^{L \times C}$ represents all categorical data, $\mathbf{N} \in \mathbb{R}^{L \times N}$ represents all continuous (numeric) data, $L$ is the total number of samples (rows), $C$ is the total number of categorical features (columns), $N$ is the total number of continuous features, and $\mathbf{X} \in \mathbb{R}^{L \times (C+N)}$ is arbitrarily defined as the concatenation of categorical and continuous data on the feature axis — a loose replacement of $\mathbf{X} \in \left( \mathbb{N}^{L \times C} \right) \times \left( \mathbb{R}^{L \times N} \right)$ — for the ease of notation for full samples. In many real-world applications, we additionally require a target array $\boldsymbol{y} \in \mathbb{A}^L$, where $\mathbb{A}$ is an arbitrary domain that varies according to the application tasks, e.g., $\boldsymbol{y} \in \{0, 1\}^L$ for binary classification such as fraud detection, and $\boldsymbol{y} \in \mathbb{R}^L$ for regression such as sales prediction. Common task of machine learning on tabular data is to achieve a predictive model $f : \mathbb{N}^C \times \mathbb{R}^N \mapsto \mathbb{A}$, such that $\hat{y}_l = f(\mathbf{X}_l)$ minimizes the difference between predictions $\hat{\boldsymbol{y}}$ and true targets $\boldsymbol{y}$ given all labeled samples $(\mathbf{X}, \boldsymbol{y})$. Popular options for $f$ are GBDTs and neural networks, as introduced in Section 1.

For every categorical feature $\mathbf{C}_{:,i}, \forall 1 \leq i \leq C$, its cardinality $S_i$ is defined as the number of unique values in $\mathbf{C}_{:,i}$, and it is always assumed that $\mathbf{C}_{:,i} \in [1, S_i]^L$. However, raw categorical data is always composed of plain text identifiers, e.g., gender, rather than integers. To process plain text into $[1, S_i]$, a corpus of unique identifiers $V_i$ is collected from the $i$-th feature during training, and each category in $V_i$ will be mapped to an integer $i$ to construct valid $\mathbf{C}_{:,i}$, e.g., "Female" to 0 and "Male" to 1. This process of collection and mapping, often referred to as *Ordinal Encoding*, ensures that all tabular data can be stored in a numeric format as defined, and is one of the most basic categorical encodings.

While in most existing studies, it is assumed that the categorical cardinality $S_i$ is a small constant, $S_i$ may grow endlessly in practice, resulting in the large or infinite cardinality categorical encoding challenge in many real-world applications, e.g., tags in recommendation systems.

The purpose of categorical encoding is to construct a translation function $E_i : \mathbb{N} \mapsto \mathbb{R}^{d_i}$ for each feature $1 \leq i \leq C$, where $d_i \in \mathbb{N}$ is an arbitrary encoding dimensionality. For example, one of the most basic and popular encodings, One-hot Encoding, is defined by $E_i^{\text{(1-hot)}} : \mathbb{N} \mapsto \{0, 1\}^{S_i}$ such that $E_i^{\text{(1-hot)}}(c) := \boldsymbol{e}^{(c)}$ where $\boldsymbol{e}_c^{(c)} = 1 \wedge \boldsymbol{e}_{k \neq c}^{(c)} = 0$ for any categorical value $c \in [1, S_i]$. Aggregating categorical encodings $[E_i]_{i=1}^C$ for all categorical features together, we construct a categorical encoding process $E : \mathbb{N}^C \times \mathbb{R}^N \mapsto \mathbb{R}^{\left(\sum_i d_i + N\right)}$ for full tabular data $\mathbf{X}$ such that

$$\forall \boldsymbol{x} = (\boldsymbol{c}, \boldsymbol{n}) \in \mathbb{N}^C \times \mathbb{R}^N, E(\boldsymbol{x}) = \left(\Big\|_{i=1}^C E_i(\boldsymbol{c}_i)\right) \Big\| \boldsymbol{n}$$

where $\|$ is the symbol for vector (array) concatenation. After applying process $E$, tabular sample $\boldsymbol{x}$ will be translated into pure numeric format $E(\boldsymbol{x})$ that any machine learning algorithm can utilize. For the ease and consistency of notation, we extend encoding definition to continuous features, and always assume that any continuous encoding $E_j : \mathbb{R} \mapsto \mathbb{R}$ is identity function $E_j(n) = n, \forall n \in \mathbb{R}$ for any continuous feature $C + 1 \leq j \leq C + N$.

## 3 RELATED WORKS

**Supervised Encodings.** As far as we know, supervised encodings are the most popular encoding techniques used in tabular learning when targets $\boldsymbol{y}$ are available. The most representative work is Target Encoding (Hayashi, 2011; Micci-Barreca, 2001), which uses the mean of all targets corresponding to the same category as the encoding. Other popular supervised variants follow the same schema but adopt different target statistics or smoothing techniques. For example, CatBoost Encoding (Prokhorenkova et al., 2018) quantizes continuous targets into buckets; Quantile Encoding (Mougan et al., 2021) uses target quantiles instead of means; James-Stein Estimator (James & Stein, 1992; Said, 2017) uses a biased mean estimation to collect target means of multiple categories jointly; GLMM (Gelman & Hill, 2007) simply learns the linear coefficient between the category and targets as the encodings. Another trend from deep learning is to make categorical encodings part of the learnable weights of neural networks (Mikolov et al., 2013), and the variants in this trend differ in their regularization techniques.

**Unsupervised Encodings.** On the other hand, in many real-world applications, target labels are not available, e.g., anomaly detection, thus unsupervised encodings are preferred. The most common unsupervised encodings are Ordinal Encoding and One-hot Encoding (Hancock & Khoshgoftaar, 2020; Potdar et al., 2017). However, they both have fatal risks: Ordinal Encoding introduces unexpected ordering between categories, which may confuse machine learning models and cause over-fitting; One-hot Encoding has a dimensionality equal to the number of categories, thus suffering from the curse of dimensionality (Hancock & Khoshgoftaar, 2020; Pargent et al., 2019; Potdar et al., 2017; Verleysen & François, 2005). To control the dimensionality, Hash Encoding (Weinberger et al., 2009) projects categories into hash buckets, and Random Encoding assigns each category a random vector of short length (Ahlswede & Zhang, 2006; Hutchinson, 1989), but they all have randomness in encoding generation, thus extrapolating poorly to unseen categories. CESAMMO (Valdez-Valenzuela et al., 2022) proposes a meaningful pruning over Random Encoding, which preserves the correlation of each category with other features; however, it needs to compute polynomial approximations between every encoding dimension and every feature, which does not scale well. Similar concepts are shared by (Kuri-Morales, 2018; 2015) but with even more severe scalability issues. As far as we know, Count Encoding, which replaces each category by its frequency statistics, is the only scalable unsupervised categorical encoding (Pargent et al., 2019). The representative work is SDV (Patki et al., 2016), which uses the cumulative distribution function as the frequency statistics and adds Gaussian noise for better inference generalization.

**Semantic Encodings.** With the recent rise of Large Language Models (LLMs) (Floridi & Chiriatti, 2020), utilizing word embeddings from LLMs to construct categorical encodings has gained increasing interest in tabular learning (Hegselmann et al., 2023; Onishi et al., 2023; Wang et al., 2023; Zhang et al., 2023). For example, TabLLM (Hegselmann et al., 2023) describes each tabular sample $\boldsymbol{x}$ in text and directly uses LLMs to make predictions from the text description. Similar concepts can also be found in conventional categorical encoding techniques, such as Similarity Encoding (Cerda et al., 2018), Min-Hash and Gamma-Poisson Encoding (Cerda & Varoquaux, 2020), or Word Embedding Encoding (Mikolov et al., 2013). They all ensure that encodings with similar text in the corpus $V_i$

are close enough to preserve semantic information. However, while these ideas have the benefit of borrowing external knowledge, they may not be suitable for highly abstracted or privacy-protected categories, e.g., anonymous categories (such as "Type-A").

# 4 CONTEMPORARY CONTINUOUS AGGREGATION

In this section, we introduce our unsupervised categorical encoding proposal: Contemporary Continuous Aggregation (CCA), which effectively and efficiently extrapolates to unseen categories without any learning. We first provide the theoretical inspiration of CCA in Section 4.1, then formally define the design and practical implementation of CCA in Section 4.2.

## 4.1 THEORETICAL INSPIRATION

As claimed in (Kuri-Morales, 2018; 2015; Valdez-Valenzuela et al., 2022), the fundamental requirement of categorical encoding is to assign each unique category value $c$ of arbitrary feature $i$ with a group of continuous values $E_i(c)$ which preserves the interdependency (correlation) of the category with all the other values. To be more instantiated, we introduce Definitions 4.1 and 4.2 for better clarity.

**Definition 4.1 (Tabular Pair Pattern)** *The Tabular Pair Pattern for an arbitrary categorical value $c$ of $i$-th categorical feature on $j$-th feature (either categorical, continuous or target) is defined by*

$$P_{i,j}(c) = \left\{\!\!\left[ [\mathbf{X}\|\boldsymbol{y}]_{l,j} \,\Big|\, \mathbf{C}_{l,i} = c, \forall l \in [1, L] \right]\!\!\right\}, \forall 1 \le i \le C, \forall 1 \le j \le C + N + 1, \forall c \in [1, S_i] \quad (1)$$

*where $\{\!\![\cdot]\!\!\}$ is the symbol for multiset, and $[\mathbf{X}\|\boldsymbol{y}]$ is tabular data $\mathbf{X}$ with potential targets $\boldsymbol{y}$ being concatenated as the last feature. In general, $P_{i,j}(c)$ is the multiset of all $j$-th feature or target values of tabular samples whose $i$-th categorical feature is $c$.*

**Definition 4.2 (Pair Pattern Distinguishable)** *A categorical encoding $E_i$ for $i$-th categorical feature is a Pair Pattern Distinguishable Encoding for $j$-th feature (or target) if*

$$P_{i,j}(c_1) \ne P_{i,j}(c_2) \implies E_i(c_1) \ne E_i(c_2), \forall c_1, c_2 \in [1, S_i]. \quad (2)$$

*Pay attention that we do not assume $E_i(c_1) \ne E_i(c_2) \implies P_{i,j}(c_1) \ne P_{i,j}(c_2)$ since equivalent categories can have different encodings due to randomness, e.g., CESAMMO (Valdez-Valenzuela et al., 2022) or SDV (Patki et al., 2016).*

The simplest categorical encoding is a **Self Pair Pattern Distinguishable** encoding, which can distinguish any self tabular pair patterns $P_{i,i}(c)$ for any categorical feature $i$. Such encodings are constructed by bijections between the original categories and their encodings, e.g., Ordinal and One-hot Encodings. They are also considered **Perfect Pair Pattern Distinguishable** encodings, as they can distinguish all tabular patterns due to their bijectivities. However, since these encodings require bijections between the original categories and encodings, they will suffer from an out-of-distribution issue when encountering unseen categories, thus extrapolating poorly on evolving systems. Another issue with self pair pattern distinguishable encodings is that they are extremely sensitive to different categories, even when those categories exhibit similar relationships with other features. For instance, if $P_{i,j}(c_1)$ and $P_{i,j}(c_2)$ are similar, we would expect their encodings to be more similar to reflect such property than any $c_3$ whose $P_{i,j}(c_3)$ differs significantly. However, self pair pattern distinguishable encoding can not guarantee this expectation, e.g., $c_1, c_2, c_3$ will have same encoding distance with each other for One-hot Encoding, and $c_3$ may even be closer to $c_1$ for Ordinal Encoding.

To overcome this sensitivity, **Target Pair Pattern Distinguishable** encoding which can distinguish any target tabular pair pattern $P_{i,C+N+1}(c)$, e.g., Target Encoding, is proposed. Since such kind of encodings utilizes the interdependency between categories and targets as the encoding, they reduce the encoding sensitivity to category difference, thus are more robust and beneficial for target predictions. While target pair pattern distinguishable encoding has successfully proved their power in many studies, they necessarily requires targets to be provided for all categories during development. However, as introduced in Section 1, many real-world scenarios do not ideally have available targets and may frequently encounter unseen categories at inference time.

Thus, it is necessary to design a categorical encoding without reliance on learning from targets that can handle unseen categories. In Section 4.2, we propose **Contemporary Continuous Aggregation (CCA)** encoding, a theoretically sound design with a practical implementation to address the challenge of unsupervised category extrapolation.

## 4.2 Design Definition

In an unsupervised scenario where categories continue to grow and no target labels are available, the only knowledge we can utilize for encoding is the correlation between each category with all the other features. To be more instantiated, we need an expressive representation for

$$[P_{i,j}(c)]_{1 \leq j \leq C+N, j \neq i}, \forall 1 \leq i \leq C, \forall c \in [1, S_i].\tag{3}$$

Pay attention that the categorial feature $i$ being encoded itself is excluded since self pair pattern distinguishable encoding is equivalent to a bijection, thus does not have the extrapolation ability to unseen categories. Furthermore, since expressive representation involving categorial features is equivalent to categorical encodings, in this design, we only consider constructing an expressive representation for continuous features

$$P_i^{(\text{cont})}(c) := [P_{i,j}(c)]_{C+1 \leq j \leq C+N}, \forall 1 \leq i \leq C, \forall c \in [1, S_i].\tag{4}$$

We refer encodings from such representations as **Continuous Pair Pattern Distinguishable** encoding.

One straightforward way to construct an expressive representation $f$ for any $P_i^{(\text{cont})}(c)$ is to construct an expressive representation $g$ for every multiset $P_{i,j}(c)$ of $P_i^{(\text{cont})}(c)$ such that

$$E_i^{(\text{CCA})}(c) := f\left(P_i^{(\text{cont})}(c)\right) = \Big\|_{j=C+1}^{C+N} g\big(P_{i,j}(c)\big), \forall 1 \leq i \leq C, \forall c \in [1, S_i].\tag{5}$$

Thus, this problem can be reduced to finding expressive representations for multisets of continuous values. Effective and efficient multiset representations have been widely studied through invariant and equivariant representation theory in graph applications such as DeepSet (Zaheer et al., 2017), Relational Pooling (Murphy et al., 2019), Set Transformer (Lee et al., 2019), and many other artifacts (Keriven & Peyré, 2019; Maron et al., 2018; Sannai et al., 2019; Puny et al., 2021). In this work, we adopt Principle Neighborhood Aggregation (PNA) (Corso et al., 2020), a theoretically sound non-parametric multiset representation for continuous values as the kernel function $g$ in Equation (5).

**Theorem 4.1 (Principle Neighborhood Aggregation (Corso et al., 2020))** *In order to discriminate between multisets of size $L$ whose underlying set is $\mathbb{R}$, at least $L$ aggregators are needed, and the moments of a multiset (Equation* (6)*) are a good practice of such aggregators.*

$$M_n(X) = \sqrt[n]{\mathbb{E}\left[\left(X - \mathbb{E}[x]\right)^n\right]}, \forall n \geq 1.\tag{6}$$

Since the expressive multiset representation defined in Theorem 4.1 can grow infinitely with size $L$, PNA proposes a constant sized approximation that empirically work nicely

$$g^{(\text{PNA})}(P) = \begin{bmatrix} 1 \\ \frac{1}{Z_1} \cdot \log\left(|P| + 1\right) \\ \frac{1}{Z_2} \cdot \frac{1}{\log(|P|+1)} \end{bmatrix} \otimes \begin{bmatrix} \mathbb{E}[P] \\ \sigma(P) \\ \min(P) \\ \max(P) \end{bmatrix}\tag{7}$$

where $|P|$ is the degree (cardinality) of multiset $P$, $Z_1, Z_2$ are coefficients which normalize related values between $[0, 1]$, $\mathbb{E}[P]$ is the mean of $P$, $\sigma(P)$ is the standard deviation of $P$, and $\otimes$ is the symbol for tensor product. The aggregators (right component) can be treated as moments of $n \in \{1, 2, \infty\}$.

The scalars (left component) with term $\log\left(|P| + 1\right)$ are proposed to guarantee generalization of the neural network proposed in PNA, but in this work, we focus only on conventional machine learning algorithms, thus Equation (7) is simplified into a form carrying similar information

$$g^{(\text{CCA})}(P) = \left[\log\left(|P| + 1\right), \frac{\text{nan}(P)}{|P| + 1}, \mathbb{E}(P), \sigma(P), \min(P), \max(P)\right]\tag{8}$$

where $\text{nan}(P)$ is the number of Not-a-Number elements (NaN) in $P$, which is often used as place-holder for missing values in real-world tabular data.

Combining Equations (5) and (8) together, we achieve our final proposal of Contemporary Continuous Aggregation (CCA) formula

$$E_i^{(\text{CCA})}(c) = \left\|_{j=C+1}^{C+N} g^{(\text{CCA})}\big(P_{i,j}(c)\big), \forall 1 \le i \le C, \forall c \in [1, S_i]. \tag{9}$$

with two advantages: First, CCA encoding only relies on raw input itself, thus can automatically extrapolate to any unseen categories without any learning; Second, the non-parametric statistics Equation (8) for each $P_{i,j}(c)$ can be simply maintained through its cardinality, minimum, maximum, sum and squared sum each of which requires only $\mathcal{O}(1)$ maintenance cost (both memory and time), thus the update cost of CCA is linear to number of (new) samples $\mathcal{O}(L)$ along with $\mathcal{O}(L \log L)$ cost for indexing all categories $c$, promising its scalability to handle large amounts of data in practice.

**Large Dimensionality Risk.** While CCA does not suffer from large dimensionality caused by high category cardinality like One-hot Encoding, its encoding dimensionality is still linear to the number of features, thus the final encoding dimensionality can be at worst quadratic to the number of raw features. In real-world applications, we may have redundantly rich tabular features that can grow up to hundreds or thousands, which poses the risk of large dimensionality for CCA. To suppress this risk, if the CCA encoding results in more than 128 dimensions, we adopt unsupervised dimensionality reduction techniques, including PCA (KPFRS, 1901) and feature agglomeration (Nielsen & Nielsen, 2016), to project it into 128 dimensions. While we consider only unsupervised methods, supervised dimensionality reduction techniques such as LDA (Fisher, 1936) are also compatible with CCA for unseen categories.

**Encode Correlation with Other Categorical Features.** While in this work, the correlation between different categorical features is excluded from the encoding, Equations (5) to (9) are compatible with categorical values. Thus, if we additionally include all categorical features **C** in the CCA generation, it will result in a perfect pair pattern distinguishable encoding, provided we do not need to discriminate between different categories that have the same correlations with all other features. The only issue is that it would inherit the risks of Ordinal Encoding, as introduced in Section 3

## 5 EXPERIMENTS

In this section, we evaluate the CCA encoding under different situations to showcase its efficacy. In Section 5.1, we introduce the adopted datasets and encoding baselines in this work. In Section 5.2, we experiment with CCA in a zero-shot transfer learning situation simulating a real-world challenge as introduced Section 1. In Sections 5.3 to 5.5, we continue to perform ablation studies for CCA to understand its behavior under various learning environments.

### 5.1 DATASETS AND ALGORITHMS

Since CCA is a categorical encoding relies on continuous features, we focuses on datasets whose features include both categorical and continuous variables. We select suitable datasets from multiple tabular data sources including UCI ML Repository (Asuncion & Newman, 2007), Kaggle (Kaggle, Google, 2010), OpenML (Vanschoren et al., 2013), AutoML (Guyon et al., 2019). A detailed summary of adopted dataset sources are provided in Table 1.

For encodings other than CCA, we consider SDV Encoding (Patki et al., 2016) as competing unsupervised baseline. Ordinal Encoding and One-hot Encoding are not considered since their encodings will generate out-of-distribution values on unseen categories, thus lacking extrapolation ability. For other unsupervised encodings covered in Section 3, they additionally suffer scalability issue in real-world applications.

We specially consider **Discard Encoding**, an unsupervised encoding that discards all categorical features and applies machine learning only on the continuous features shared between training and test sets. We adopt this trivial encoding to study the importance of categorical features and depict the power of CCA accordingly: If Discard Encoding performs poorly, it means that the continuous features are insufficient for the prediction task, thus categorical values and encoding techniques are

Table 1: **Dataset Statistics.** For the study of CCA performance, we collect essential dataset properties including numbers of categorical and continuous features, maximum and total cardinalities over all categorical features and target label imbalance rates (positive rate w.r.t. negative). We also provide the link to each public dataset source for ease of reproducibility.

| Dataset | Categorical | Continuous | Max Cardinality | Total Cardinality | $^{\#Positive}/_{\#Negative}$ | Source |
|---|---|---|---|---|---|---|
| Vehicle Claims [1] | 13 | 4 | 86,327 | 100,903 | 0.102 | Github |
| Vehicle Insurance | 24 | 4 | 1782 | 153 | 0.975 | Github |
| Insurance Claims | 17 | 16 | 39 | 1,897 | 0.313 | Kaggle |
| Shrutime | 6 | 5 | 2,932 | 2,952 | 0.261 | OpenML |
| Census Income | 8 | 6 | 42 | 102 | 0.318 | UCI ML |
| Purchase Intention | 10 | 7 | 311 | 420 | 0.186 | UCI ML |
| Blastchar | 17 | 2 | 6,531 | 6,574 | 0.357 | Kaggle |
| Jasmine | 136 | 8 | 2 | 272 | 0.977 | AutoML |
| Obesity Levels | 8 | 8 | 7 | 28 | 0.155 | UCI ML |
| Telecom | 3 | 10 | 2 | 6 | 0.186 | UCI ML |
| Seismic Bumps | 4 | 14 | 3 | 10 | 0.069 | UCI ML |
| Predictive Maintenance | 1 | 5 | 3 | 3 | 0.034 | UCI ML |
| Bank Marketing | 10 | 10 | 12 | 53 | 0.126 | UCI ML |
| QSAR Biodegradation | 5 | 36 | 11 | 21 | 0.510 | UCI ML |

critical for machine learning, and we expect CCA to perform better on this task. Otherwise, if Discard Encoding performs reasonably well, categorical features are not essential for the prediction, and the CCA encoding may introduce too many redundant dimensions, which can easily cause over-fitting. In such cases, we do not expect CCA to be the best encoding.

In the supervised ablation study, where we have available target labels during training, we also consider CatBoost Encoding, a supervised target encoding embedded in the CatBoost algorithm that is widely adopted in industry, as an auxiliary baseline. The purpose of this baseline is to show that CCA can achieve similar or even better performance without relying on target statistics.

For tabular learning frameworks, we adopt the representative state-of-the-art GBDT methods including XGBoost, CatBoost and LightGBM for zero-shot transfer learning experiment and supervised ablation study. We do not consider deep neural networks since they lack explainability, scale poorly and only achieve similar performance as GBDTs with more costs (Grinsztajn et al., 2022; Shwartz-Ziv & Armon, 2022), thus are not widely adopted in industry. In unsupervised ablation study where no targets are presented, we consider unsupervised learning approaches instead including One-Class SVM (Schölkopf et al., 2001; Ruff et al., 2018) [1], Isolation Forest (Liu et al., 2008) and Local Outlier Factor (Breunig et al., 2000).

In all considered datasets, positive labels are (mostly far) less than negative labels. Thus, we select **Precision-Recall Area Under Curve (PR-AUC)** as the evaluation metric for all experiments which is insensitive to label imbalance. Furthermore, we sort all datasets by the descending order of categorical feature importance (the mean difference from other encodings to Discard Encoding). Consequently, earlier datasets rely more on categorical encodings, and are more valuable for encoding comparison. For the test performance comparison, we select the best model and hyperparameter configuration based on validation PR-AUC for each encoding separately. Hyperparameter details are provided in Appendix A.

## 5.2 ZERO-SHOT TRANSFER LEARNING

Our CCA encoding is primarily designed to transfer a fraud detection system between two regions with category shifting challenge without any labeled data on targeting region. However, due to confidential reasons, we are not able to make this application dataset public, therefore we design a synthetic process to simulate the same task on public datasets.

**Synthesis Process.** The challenge of our task is that categories may be renamed, merged into new categories, or splitted into sub-categories, thus to reproduce such situations, we uniformly select

---

[1]The *Vehicle Claims* dataset is large, causing some algorithms, e.g., One-Class SVM, to scale poorly. For such cases, we adopt scalable approximations instead of the original algorithms, e.g., the Stochastic Gradient Descent (SGD) approximation for One-Class SVM with a Gaussian kernel.

Table 2: **Transfer Learning.** CCA achieves the best overall performance in zero-shot transfer learning scenarios. On the datasets where categorical features are important, CCA performs the best; while in other cases, CCA still achieves close-to-top performance.

Table 3: **Supervised Learning.** In the supervised ablation study, CCA outperforms target encoding in all cases except for "Predictive Maintenance". CCA is always the best when categorical features are important, and it maintains performance close to the top in the remaining cases.

| Dataset | Disacrd | SDV | __CCA__ |
|---|---|---|---|
| Vehicle Claims [1] | 0.186 | 0.348 | **0.935** |
| Vehicle Insurance | 0.486 | 0.559 | **0.650** |
| Insurance Claims | 0.291 | 0.379 | **0.478** |
| Shrutime | 0.599 | 0.674 | **0.676** |
| Census Income | 0.719 | 0.787 | **0.795** |
| Purchase Intention | 0.713 | 0.747 | **0.752** |
| Blastchar | 0.611 | 0.631 | **0.643** |
| Jasmine | 0.723 | 0.735 | **0.751** |
| Obesity Levels | 0.897 | 0.865 | **0.959** |
| Telecom | 0.879 | 0.860 | **0.920** |
| Seismic Bumps | 0.180 | 0.154 | **0.203** |
| Predictive Maintenance | **0.625** | 0.595 | 0.614 |
| Bank Marketing | **0.661** | 0.623 | 0.653 |
| QSAR Biodegradation | **0.892** | 0.869 | 0.651 |
| **Mean Performance** | 0.604 | 0.630 | **0.691** |

| Dataset | Discard | SDV | __CCA__ | CatBoost |
|---|---|---|---|---|
| Vehicle Claims [1] | 0.184 | 0.998 | **1.000** | **1.000** |
| Vehicle Insurance | 0.508 | 0.795 | **0.838** | 0.820 |
| Insurance Claims | 0.358 | 0.613 | **0.786** | 0.700 |
| Shrutime | 0.604 | 0.656 | **0.680** | 0.558 |
| Census Income | 0.711 | 0.793 | **0.804** | 0.801 |
| Purchase Intention | 0.738 | 0.730 | **0.756** | 0.750 |
| Blastchar | 0.575 | 0.576 | **0.608** | 0.581 |
| Jasmine | 0.718 | 0.734 | **0.772** | 0.755 |
| Obesity Levels | 0.917 | 0.890 | **0.948** | 0.910 |
| Telecom | 0.908 | 0.932 | **0.962** | 0.931 |
| Seismic Bumps | 0.184 | 0.151 | **0.211** | 0.158 |
| Predictive Maintenance | 0.663 | 0.682 | 0.659 | **0.705** |
| Bank Marketing | **0.657** | 0.645 | 0.653 | 0.651 |
| QSAR Biodegradation | 0.917 | 0.889 | **0.921** | 0.908 |
| **Mean Performance** | 0.617 | 0.720 | **0.757** | 0.731 |

categories from public datasets at test stage, and perform one of the following corruptions: Either directly change the selected one to a totally new category that never appears in training or split one third of its samples into a new category, and remaining two thirds into another new category.

As we can observe from Table 2, CCA encoding achieves a clearly better overall performance on zero-shot transfer learning showcasing its impressive unsupervised extrapolation ability. If we split all datasets based on if categorical features are fatal or not, i.e., Discard Encoding outperforms one of the other encodings, we can notice that CCA encoding is always the best technique in the first half where categorical encodings are critial for predictions, empirically justify its power as categorical encoding. On the other half where categorical features are not important, CCA is no longer the best, but still maintains close-to-best performance if not. The reason behind such degradation is over-fitting due to redundant sparsity on encoded $E(\mathbf{X})$ (Verleysen & François, 2005).

To conclude, Table 2 shows that CCA is an extraordinary unsupervised categorical encoding for new category extrapolation when categorical features are critical, and is still a powerful encoding toolkit even when categorical encoding is not such important.

## 5.3 SUPERVISED LEARNING

While CCA has proved its extrapolation power on zero-shot transfer learning, it is still important to study its behavior on common scenarios to make the conclusion more persuasive. Thus, we continue to conduct supervised experiments on the same datasets in Table 3. As we can observe, the supervised performance is mostly consistent with the zero-shot transfer results: CCA achieves the best overall performance; CCA is always the best when categorical features are important; and CCA achieves close-to-top performance when categorical features are not as important. Furthermore, in all cases except "Predictive Maintenance", CCA consistently outperforms target encoding. We believe the failure reason on that dataset is over-fitting caused by the extreme sparsity on categorical features (see Table 1).

To conclude, Table 3 empirically showcases that CCA, as an unsupervised categorical encoding, is a potential competitor to target encoding even under supervised learning.

## 5.4 UNSUPERVISED LEARNING

Since positive labels are (mostly far) fewer than negative labels on all datasets, we continue to study encoding performance under unsupervised scenario through anomaly detection. The result is illustrated in Table 4: While CCA achieves the best overall performance, the per dataset performance

is mixed compared to SDV. The reason is that SDV is good at finding low frequency category outliers while CCA is not since aggregation statistics other than degree may potentially suppress the outstanding low frequency; On the other hand, CCA is good at find bizarre correlation patterns that SDV is incapable detecting.

To conclude, SDV and CCA have their own advantages in unsupervised scenarios, thus need careful study per dataset to decide which one to choose.

### 5.5 Statistics Aggregation Contribution

We continue to study encoding statistics importance in Table 5: We split 6 collected statistics of Equation (8) into three kinds: The first two as degree statistics, the middle two as distribution statistics, and the last two as range statistics, then compare their relative mean degradation w.r.t. the best performance.

As we can see in Table 5, using only range statistics for CCA encoding construction degrades the most on overall performance, then degree statistics degrades the second, and distribution statistics degrades the least. This order is reasonable since distribution statistics is relatively more informative than the other two kinds in representing correlations, thus best fits the theoretical inspiration of CCA. On the other hand, the degree statistics carries equivalent information as SDV baseline, thus it is expected for it to work closely as full statistics. Finally, range information solely is the least useful since it may use extreme outliers to generate emebeddings, resulting in poor generalization performance.

Table 4: **Unsupervised Learning.** On unsupervised fraud detection, while CCA achieves the best overall performance, the per dataset performance is mixed with SDV baseline.

| Dataset | Discard | SDV | **CCA** |
|---|---|---|---|
| Vehicle Claims [1] | 0.093 | 0.096 | **0.097** |
| Vehicle Insurance | 0.483 | **0.529** | 0.508 |
| Insurance Claims | 0.303 | 0.293 | **0.333** |
| Shrutime | 0.221 | **0.226** | 0.217 |
| Census Income | 0.234 | 0.233 | **0.236** |
| Purchase Intention | 0.160 | 0.165 | **0.173** |
| Blastchar | 0.250 | **0.254** | 0.253 |
| Jasmine | 0.474 | **0.503** | 0.502 |
| Obesity Levels | 0.174 | 0.252 | **0.280** |
| Telecom | 0.168 | **0.224** | 0.206 |
| Seismic Bumps | **0.069** | **0.069** | 0.067 |
| Predictive Maintenance | 0.054 | 0.050 | **0.073** |
| Bank Marketing | **0.146** | 0.129 | 0.144 |
| QSAR Biodegradation | 0.326 | **0.330** | **0.330** |
| **Mean Performance** | 0.225 | 0.240 | **0.244** |

Table 5: **Aggregation Statistics Importance.** Using only distribution statistics degrades the least in overall performance, followed by degree and range statistics.

| Aggregations | Transfer | Supervised | Unsupervised | Mean |
|---|---|---|---|---|
| Degree | −13.88% | −1.90% | −5.77% | −7.18% |
| Distribution | −13.70% | −1.99% | −5.17% | −6.95% |
| Range | −22.21% | −4.72% | −5.09% | −10.67% |

## 6 Conclusion

This work proposes Contemporary Continuous Aggregation (CCA), a novel categorical encoding that can effectively and efficiently extrapolate to unseen categories for tabular learning. We first proved the expressivity of CCA's inspiration and implementation through representation theory. Then, we empirically showed that CCA outperforms preceding encodings in a zero-shot transfer learning challenge from a real-world application and its simulation on public datasets. Finally, we compared CCA's performance on regular supervised and unsupervised tasks, showcasing that it achieves similar or even better performance when compared to other encodings. Thereby, we demonstrate that CCA is a worthy addition with extrapolation power to the categorical encoding toolbox for tabular learning.

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

## A HYPERPARAMETER

For each encoding experiment, we perform a grid hyperparameter search to find the best model and configuration based on the PR-AUC score on the validation data. Some datasets have a separate validation dataset defined, while others do not. For those datasets without a predefined validation set, we split the tuning data into training and validation sets in a $7 : 1$ proportion. If even the test data is not defined, we split the entire dataset into training, validation, and test sets in a $7 : 1 : 2$ proportion. For zero-shot transfer learning and supervised learning experiments, we consider XGBoost, CatBoost, and LightGBM models. For unsupervised learning experiments, we consider One-Class SVM, Isolation Forest, and Local Outlier Factor models. In the following sections, we provide the searching configurations for each model separately.

**XGBoost.** We consider maximum number of leaves in each decision tree [8, 16, 32, 64], maximum depth of each decision tree [3, 6, 9, 12], learning rates $\left[5 \times 10^{-4}, 5 \times 10^{-3}, 5 \times 10^{-2}\right]$, and maximum number of ensembling decision trees (iterations) [10, 30, 60, 100]. All the other configurations keep as default.

**CatBoost.** We consider maximum depth of each decision tree [3, 6, 9, 12], learning rates $\left[5 \times 10^{-4}, 5 \times 10^{-3}, 5 \times 10^{-2}\right]$, and maximum number of ensembling decision trees (iterations) [10, 30, 60, 100]. All the other configurations keep as default.

**LightGBM.** We consider maximum number of leaves in each decision tree [8, 16, 32, 64], maximum depth of each decision tree [3, 6, 9, 12], learning rates $\left[5 \times 10^{-4}, 5 \times 10^{-3}, 5 \times 10^{-2}\right]$, and maximum number of ensembling decision trees (iterations) [10, 30, 60, 100]. All the other configurations keep as default.

**One-Class SVM.** We consider outlier fraction [0.1, 0.15, 0.2], and kernel [RBF, Polynomial, Sigmoid, Linear]. All the other configurations keep as default.

**Isolation Forest.** We consider outlier fraction [0.1, 0.15, 0.2, Automatic], maximum number of decision trees (iterations) [10, 30, 60, 100] and maximum number of samples used to build each decision tree [Full, Automatic]. All the other configurations keep as default.

**Local Outlier Factor.** We consider outlier fraction [0.1, 0.15, 0.2, Automatic], maximum number of closest neighbors being treated as similar [10, 30, 60, 100] and maximum number of leaves in each decision tree [8, 16, 32, 64]. All the other configurations keep as default.

