# OpenReview forum: "Contemporary Continuous Aggregation: A Robust Categorical Encoding for Zero-Shot Transfer Learning on Tabular Data"
_ICLR.cc/2025/Conference — ICLR 2025 Conference Withdrawn Submission_

### Official Review · Reviewer_eeJu · 2024-10-16

**Soundness:** 2
**Presentation:** 2
**Contribution:** 2
**Rating:** 3
**Confidence:** 4

**Summary:**

The paper tackles the issue of managing unseen categorical variables during test time in tabular learning by introducing Contemporary Continuous Aggregation (CCA). CCA presents a novel and theoretically sound approach to categorical encoding, which can generalize to unseen categories without the need for additional training. The method depends only on statistics from raw input data, allowing it to scale efficiently with low computational and memory costs—an important consideration for real-time applications with high-volume workloads.

**Strengths:**

- The problem addressed by the paper is both industrially relevant and practically significant.
- The paper and the proposed method are clearly presented and easy to understand.
- Empirical results demonstrate that CCA outperforms existing encoding techniques in unsupervised extrapolation to unseen categories.
- The submission of code to ensure reproducibility is commendable.

**Weaknesses:**

**Methodology**
1. **Moment Estimation for New Categorical Variables**
   When introducing new categorical variables, it is essential to estimate the first, second, and higher-order moments (mean, standard deviation, minimum, and maximum) of all continuous variables associated with the new categorical variable. However, the paper seems to assume that the test set is provided offline in its entirety, calculating these moments accordingly. In an online setting, where the number of samples for newly introduced categorical variables is likely to be limited, this estimation could be highly inaccurate. The authors should provide a more detailed explanation of how their method can accurately estimate these moments in an online scenario where the test set is introduced sequentially and may be small.

2. **Theoretical Analysis of Theorem 4.1**
   While Theorem 4.1 presents a theoretical analysis, it is derived from prior work (Corso et al., 2020), offering no theoretical novelty. Additionally, the discussion lacks depth in addressing the practical implications of the theorem. The authors could enhance this section by discussing how their findings extend or build upon the existing literature.

3. **Versatility of the Proposed Method**
   The proposed method's applicability is limited in the absence of continuous columns. In the section titled **Encode Correlation with Other Categorical Features**, the authors suggest using ordinal encoding for categorical features. However, the mean, standard deviation, minimum, and maximum values become meaningless in this context, which undermines the validity of the proposed approach. An empirical verification should be conducted to assess the effectiveness of this method when applied in practice.

**Experiments**
1. **Incremental Test Stream Evaluation**
   The experiments should be conducted in an environment where the test stream is introduced incrementally and online. Furthermore, evaluating the method's performance in scenarios where target categories increase over time, as is typical in online settings, is crucial.

2. **Evaluation Metrics**
   The evaluation metric is currently limited to AUPRC. To strengthen the paper's claims, the authors should also report additional metrics such as AUROC, accuracy, and F1 Score.

3. **Baseline Comparisons**
   The number of baseline methods is insufficient. Even if scalability issues prevent certain baselines from being evaluated on larger datasets, the authors should have tested them on smaller datasets. Presenting the performance of methods like Ordinal Encoding or One-Hot Encoding, even if they perform poorly, would provide valuable context for the results.

4. **Dataset Variety**
   The number of datasets utilized in the experiments is too small. Current best practices suggest evaluating methods across nearly 100 datasets for comprehensive validation purposes.

5. **Scope of Downstream Tasks**
   The focus on binary classification (e.g., anomaly detection) is a limitation. The authors should consider testing the method on multi-class classification and regression tasks to showcase its versatility.

6. **Validation on Neural Networks**
   Although gradient-boosted decision trees (GBDTs) are commonly used in industry, it is important from a research perspective to validate the method on deep neural networks, particularly large language models (LLMs), to demonstrate broader applicability.

7. **Marginal Improvement**
   In Table 4, the proposed method shows only a 0.4% improvement over SDV, which is marginal. The authors should provide a more substantial analysis explaining the significance of this improvement.

8. **Empirical Analysis of Performance Gains**
   The paper lacks sufficient empirical analysis to explain why the proposed method outperforms others. For example, it does not clarify the specific reasons behind the performance gains achieved through the proposed categorical encoding method.

[1] Yan et al. "Making Pre-trained Language Models Great on Tabular Prediction." ICLR 2024.

**Questions:**

- Which normalization function was used for continuous variables?

---

### Official Review · Reviewer_S3ey · 2024-10-31

**Soundness:** 3
**Presentation:** 1
**Contribution:** 3
**Rating:** 5
**Confidence:** 4

**Summary:**

The paper presents Contemporary Continuous Aggregation (CCA), a categorical encoding method designed to handle unseen categories in tabular data. CCA treats categorical values as multisets of other feature values associated with each category, allowing it to capture relationships between a given category and other features without requiring re-training. The multiset representation is generated using statistical measures (e.g., mean, standard deviation), enabling CCA to extrapolate effectively to unseen categories in zero-shot transfer learning contexts. The experimental section compares CCA with other encoding techniques across several datasets, and the authors claim that CCA achieves favorable results in both supervised and unsupervised learning tasks.

**Strengths:**

- Relevance of Problem Addressed: The challenge of handling unseen categories is a significant issue in tabular data, especially for dynamic applications. The problem formulation aligns well with real-world needs, making the research highly relevant.

- Sound Encoding Methodology: By representing each category’s interaction with other features through statistical aggregates, the CCA encoding is logically designed for efficient extrapolation, a particularly valuable attribute when retraining is resource-intensive or impractical.

**Weaknesses:**

- Poor Presentation and Clarity: Section 4, which contains the core methodology, is challenging to follow. The presentation of key definitions, such as the multiset encoding process, lacks clarity and could benefit from better-organized equations and explanations.

- Limited Original Theoretical Contribution: Although the paper claims to provide a theoretical basis for CCA, much of the theoretical groundwork appears to be based on the Principle Neighborhood Aggregation (PNA) framework from prior work. This reliance limits the novelty of the theoretical contributions claimed by the authors.

- Insufficient Experimental Validation: The experimental section lacks breadth in evaluating CCA across diverse types of tabular data, as only 14 datasets are used. Additionally, comparisons with baseline methods are limited, as various encoding methods (e.g., ordinal and one-hot encoding with an "unknown" class) are not tested.

- Lack of In-Depth Analysis of Encoding Properties: The paper would benefit from a more detailed analysis of the unseen category encoding itself, beyond simple performance comparisons. A deeper examination of the encoding's inherent properties and theoretical soundness would help validate the method’s utility and justify its application to unseen categories.

- Synthetic-Only Experiments for Unseen Categories: While unseen categories commonly occur in real-world datasets, experiments for zero-shot transfer were conducted only in synthetic environments, which limits the real-world applicability and validity of the results.

**Questions:**

- Could the authors clarify which backbone architecture was used in the experiments, particularly for the supervised settings?

- In the Appendix, the GBDT iterations were limited to a low number. Given that 10,000 or more iterations are typically recommended, what was the rationale for this setting?

- If all the issues mentioned in the weaknesses and questions sections are resolved, I am willing to raise my score.

---

### Official Review · Reviewer_1qot · 2024-11-01

**Soundness:** 2
**Presentation:** 3
**Contribution:** 2
**Rating:** 5
**Confidence:** 3

**Summary:**

The authors propose a new categorical feature encoding algorithm. The algorithm uses target encoding but on the remaining numerical input features. This reduces reliance on the targets and allows handling of unseen categories.

**Strengths:**

- The authors propose a compelling categorical feature encoding algorithm based on using the feature statistics correlated with individual features and principle neighborhood aggregation.

- The authors accurately described the trade-offs of their work including improved performance, ability to handle NaN features, label agnostic, and quadratic feature count scalability. CCA uses standard feature selection algorithms to reduce cardinality.

- The authors show experiments proving they outperform Catboost and standard target encoding algorithms.

**Weaknesses:**

- While I agree with the authors that GBDTs are the best baseline to compare with, I believe results on deep learning algorithms can better contextualize the applicability of CCA. Because GBDTs naturally scale better to larger feature counts, I worry whether such performance gains are also applicable to deep learning algorithms, specifically given the rise of powerful ICL tabular learning algorithms[1,2,3,4,5].

- "Ordinal Encoding and One-hot Encoding are not considered since their encodings will generate out-of-distribution values on unseen categories, thus lacking extrapolation ability." What if you take the average ordinal encoding or the average one-hot encoding in such situations?

- The method is motivated by its ability to handle out-of-distribution data, but there is not any studies directly measuring CCA's scalability and sensitivity to out-of-distribution or NaN features.

- Is there any reason why you did not test on standard benchmarks such as [1]?

[1] When Do Neural Nets Outperform Boosted Trees on Tabular Data?

**Questions:**

See weaknesses.

If the dataset only includes categorical features, the technique no longer works. Is this correct?

What are the NaN dataset statistics?

---

### Official Review · Reviewer_KRHV · 2024-11-04

**Soundness:** 3
**Presentation:** 3
**Contribution:** 3
**Rating:** 6
**Confidence:** 2

**Summary:**

This paper targets an interesting avenue of addressing categorical encoding extrapolation to unseen categories. As a fundamental principle, categorical data is often encoded using a trained encoder-decoder setup on the underlying data, however the authors rightfully suggest that in practice, data categories may change and it might not alway be convenient to run re-training to address this data drift.
The authors introduce Contemporary Continuous Aggregation (CCA) as a method for encoding categorical data that can generalize to unseen categories without retraining. The method exhibits highly scalable characteristics in time and memory and appears to work well on both unseen extrapolation and standard unsupervised settings. The method not only excels at handling unseen categories but also performs as well as or better than traditional encoding methods in normal scenarios, making it particularly valuable for real-world industrial applications.

**Strengths:**

This paper provides compelling evidence to support the importance of handling unseen categorical data encoding. Empirically the evaluate their results a large selection of comprehensive and common tabular datasets. I find the key contribution is that this method can perform on par for tasks without extrapolation, however can show superior performance in the case when extrapolation is needed. This, from a user perspective, provides a clear advantage over current state of the art and may attract more wide scale adoption. I find that the proposed theoretical approach differs quite significantly from previous encoding strategies I've encountered in Tabular Synthetic Data, such as VAE's with re-parameterization, or transformer tokenizers. I think the novelty of this paper is clear in it's unconventional approach to addressing data drift, and in particular the expandability of the encoding process to unseen categorical features.

**Weaknesses:**

I am somewhat unfamiliar with the current state of the art in this sub-field, I yeild those concerns to other reviewers. I found the preliminaries/method section to be a bit difficult to parse, especially with the mixture of equations/relations and text, I would urge the authors to consider a more simplistic presence in the main paper and divert the more lower level technicalities into the supplementary. I additionally noticed the absence of figures throughout this manuscript. I personally believe that having a methodology figure could help better illustrate the contribution/motivation, I would recommend that the authors consider this entering the discussion phase -- a simple diagram could include an existing dataset, with an avenue for more incoming categories, and how CCA can expand it's encoding generalizability to support this setting. I think this would significantly help in the readability and understanding for a larger audience.

**Questions:**

One interesting comparison I would like to see is how the CCA approach could be integrated with existing encoding/tabular synthesis techniques such as ICLR'24 TabSyn (Score based Tabular Diffusion). With the rise in creating synthetic datasets, it is quite natural to assume industry use cases where additional categories may be introduced over time -- essentially simulated with continual learning. It would be interesting to see how CCA could perform in this setting as the encoding medium.

---

### Note · Authors · 2024-11-18

**Comment:**

Thanks for the valuable reviews.

**Withdrawal Confirmation:**

I have read and agree with the venue's withdrawal policy on behalf of myself and my co-authors.